# Spectra of Self-Similar Measures

**DOI:** 10.3390/e24081142

**Published:** 2022-08-17

**Authors:** Yong-Shen Cao, Qi-Rong Deng, Ming-Tian Li

**Affiliations:** 1Center for Applied Mathematics of Fujian Province, Fujian Key Laboratory of Mathematical Analysis and Applications (FJKLMAA), School of Mathematics and Statistics, Fujian Normal University, Fuzhou 350117, China; 2School of Computing and Information Science, Fuzhou Institute of Technology, Fuzhou 350506, China

**Keywords:** spectrality, tree structure, self-similar measure, orthogonal basis, 42C05, 42A65, 28A78, 28A80

## Abstract

This paper is devoted to the characterization of spectrum candidates with a new tree structure to be the spectra of a spectral self-similar measure $\mu_{N,D}$ generated by the finite integer digit set *D* and the compression ratio $N^{-1}$. The tree structure is introduced with the language of symbolic space and widens the field of spectrum candidates. The spectrum candidate considered by Łaba and Wang is a set with a special tree structure. After showing a new criterion for the spectrum candidate with a tree structure to be a spectrum of $\mu_{N,D}$, three sufficient and necessary conditions for the spectrum candidate with a tree structure to be a spectrum of $\mu_{N,D}$ were obtained. This result extends the conclusion of Łaba and Wang. As an application, an example of spectrum candidate Λ(N,B) with the tree structure associated with a self-similar measure is given. By our results, we obtain that Λ(N,B) is a spectrum of the self-similar measure. However, neither the method of Łaba and Wang nor that of Strichartz is applicable to the set Λ(N,B).

## 1. Introduction

Let μ be a probability measure on $R^{d}$ with compact support *K*. We say that μ is a spectral measure if there exists a countable set $\Lambda \subset R^{d}$ such that the set of exponential functions $E_{\Lambda }:=\left\{\exp2\pi i\langle \lambda ,x\rangle :\lambda \in \Lambda \right\}$ is an orthogonal basis of $L^{2}\left(\mu \right)$. In this case, Λ is called a spectrum of μ and (μ,Λ) is called a spectral pair. In particular, if μ is the normalized Lebesgue measure restricted on *K*, we say *K* is a spectral set.

In [1], Fuglede introduced the notion of a spectral set in the study of the extendability of the commuting partial differential operators and raised the famous conjecture: *K* is a spectral set if and only if *K* is a translational tile. Although the conjecture was finally disproven for the case that $K\subset R^{d}$ with d≥3 and is still open for $R^{d}$ with d≤2, it has led to the development of harmonic analysis, operator theory, tiling theory, convex geometry, etc.

In 1998, Jorgensen and Pedersen [2] discovered the first singular, non-atomic spectral measure—the middle-forth Cantor measure—and proved the middle-third Cantor measure is not a spectral measure. Following this discovery, there has been much research on the spectrality of self-similar (or self-affine) measures and Moran-type self-similar (or self-affine) measures (see for example [3,4,5,6,7,8,9,10,11,12,13,14,15,16,17,18,19,20,21] and the references therein).

Consider the iterated function system (IFS) ${\left\{\phi_{j}\right\}}_{j=1}^{q}$ given by
$$\phi_{j}\left(x\right)=\frac{1}{N}\left(x+d_{j}\right),$$
where *N* is an integer with |N|>1 and $D={\left\{d_{j}\right\}}_{j=1}^{q}$ is a finite subset of R. It is well known (see [22] or [23]) that there exists a unique probability measure $\mu_{N,D}$ satisfying
$$\mu_{N,D}\left(E\right)=\frac{1}{q}\sum_{j=1}^{q}\mu_{N,D}\left(\phi_{j}^{-1}\left(E\right)\right),\;\;\mathrm{for}\;\mathrm{Borel}\;\mathrm{set}\;E\;\mathrm{of}\;R.$$
The measure $\mu_{N,D}$ is called the self-similar measure of the IFS ${\left\{\phi_{j}\right\}}_{j=1}^{q}$ and is supported on the set
$$T\left(N,D\right)=\left\{\sum_{k=1}^{\infty }d_{k}N^{-k}:d_{k}\in D,k\geq 1\right\},$$
which is the attractor of ${\left\{\phi_{j}\right\}}_{j=1}^{q}$. Given a finite set S⊂Z with ♯S=♯D, we say $\left(\frac{1}{N}D,S\right)$ is a compatible pair if the matrix ${\left[\frac{1}{\sqrt{q}}\exp\left(2\pi i\frac{d}{N}s\right)\right]}_{d\in D,s\in S}$ is a unitary matrix. In other words, $\left(\delta_{\frac{1}{N}D},S\right)$ is a spectral pair. For a finite set *A* in R,
$$\delta_{A}:=\frac{1}{♯A}\sum_{a\in A}\delta_{a},$$
where $\delta_{a}$ is the Dirac measure at *a*. Write
$$\Lambda \left(N,S\right)=\left\{\sum_{j=0}^{k}s_{j}N^{j}:k\geq 0,s_{j}\in S\right\}.$$
Using the dominated convergence theorem, Strichartz [24] proved that $\mu_{N,D}$ is a spectral measure with a spectrum Λ(N,S) under the conditions that $\left(\delta_{\frac{1}{N}D},S\right)$ is a spectral pair with 0∈S and the Fourier transform of $\delta_{\frac{1}{N}D}$ does not vanish on T(N,S). By using the Ruelle transfer operator, Łaba and Wang in [3] removed the condition that the Fourier transform of $\delta_{\frac{1}{N}D}$ does not vanish on T(N,S). Furthermore, they obtained the following conclusion:

**Theorem** **1.**(Łaba and Wang). *Let N∈N with |N|>1, D⊂Z with 0∈D, and gcd(D)=1,0∈S⊂Z. If $\left(\frac{1}{N}D,S\right)$ is a compatible pair, then $\left(\mu_{N,D},\Lambda \left(N,S\right)\right)$ is not a spectral pair if and only if there exist integers m⩾1,${\left\{s_{j}\right\}}_{j=0}^{m-1}\subset S$ and ${\left\{\eta_{j}\right\}}_{j=0}^{m-1}\subset Z\backslash \left\{0\right\}$ such $\eta_{j+1}=N^{-1}\left(\eta_{j}+s_{j}\right)$ for 0⩽j⩽m−1, where $\eta_{m}:=\eta_{0},s_{m}:=s_{0}.$*

It is well known that to prove the spectrality of the invariant measure $\mu_{N,D}$, the first key step is to construct a suitable spectrum candidate. In this process, the set $\Lambda \left(N,S\right)=S+NS+N^{2}S+\cdots$ (finite sum) is the natural spectrum candidate to be considered. Form Theorem 1, we conclude that Λ(N,S) is not a spectrum of $\mu_{N,D}$ if and only if there is a periodic orbit ${\left\{\eta_{j}\right\}}_{j=0}^{m-1}\subset Z\backslash \left\{0\right\}$ under the dual IFS $\{\psi_{i}\left(x\right)=\frac{1}{N}\left(x+s_{i}\right):s_{i}\in S\}.$ The following example implies that the natural spectrum candidate has a weak point. When D={0,1}, the invariant measure $\mu_{2,D}$ is just the Lebesgue measure on the unit interval with the unique spectrum Z. However, Λ(2,{0,1})=N≠Z in this case. In other words, the natural candidate Λ(2,{0,1}) is not a spectrum of $\mu_{2,D}$. Actually, any set with form $S+2S+2^{2}S+\cdots$ (finite sum) is not a spectrum of $\mu_{2,D}$. In this case, one needs to consider the spectrum candidate with a more general form $S_{1}+NS_{2}+N^{2}S_{3}+\cdots$ (finite sum), where $\left(\frac{1}{N}D,\;S_{i}\right)$ are compatible pairs. Moreover, it is well known that a spectral self-similar (or self-affine) measure has more than one spectrum in general. The results in [7,9,10,11] show that one may consider spectrum candidates with a tree structure. It is worth mentioning that Li [16] obtained a simplified form of Theorem 1. To the best of our understanding, partial results have been obtained in the case of a higher-dimensional space. Developing the method in [3], Dutkay and Jorgensen [14] obtained a sufficient condition for the spectral pair of self-affine measures, and Li [19] obtained a necessary condition for the natural spectrum candidate to be a spectrum of a self-affine measure.

Motivated by the above results, we considered a class of spectrum candidates with a tree structure (defined in Section 2) and obtained three necessary and sufficient conditions for such spectrum candidates not to be the spectra of $\mu_{N,D}$ (Theorem Equation 2), which generalizes Łaba and Wang’s result.

The most difficult part of the proof of Theorem Equation 2 is that the first statement implies the second. For this purpose, we show a new criterion for Λ to be a spectrum of $\mu_{N,D}$. As an application, we give an example involving a self-similar measure μ and a spectrum candidate Λ(N,B) with a tree structure in Section 4. By Theorem Equation 2, we obtain (μ,Λ(N,B)) is a spectral pair. However, neither the criterion of Łaba and Wang (Theorem 1) nor that of Strichartz [24] is applicable to this set Λ(N,B).

## 2. Preliminaries

In this section, we shall recall some basic properties of spectral measures and introduce the tree structure using symbolic space.

Let μ be a probability measure on R. The Fourier transform of μ is defined by
$$\hat{\mu }\left(\xi \right)=\int e^{-2\pi i\xi x}\;d\mu \left(x\right),\;\;x\in R.$$
We write $Z\left(\hat{\mu }\right)=\left\{\xi :\hat{\mu }\left(\xi \right)=0\right\}$. For a discrete set Λ⊂R, write $E_{\Lambda }=\left\{\exp\left(2\pi ix\lambda \right):\lambda \in \Lambda \right\}$ for a family of exponential functions in $L^{2}\left(\mu \right)$. Then, $E_{\Lambda }$ is an orthogonal family of $L^{2}\left(\mu \right)$ if and only if
$$\Lambda -\Lambda \subset Z\left(\hat{\mu }\right)\cup \left\{0\right\}.$$

Define
$$Q_{\Lambda }\left(\xi \right)=\sum_{\lambda \in \Lambda }{\left|\hat{\mu }\left(\lambda +\xi \right)\right|}^{2},\;x\in R.$$
By using the Parseval identity, Jorgenson and Pederson ([2]) obtained the following basic criterion for the orthogonality of $E_{\Lambda }$ in $L^{2}\left(\mu \right)$.

**Proposition** **1.**
*The exponential function set $E_{\Lambda }$ is an orthogonal set of $L^{2}\left(\mu \right)$ if and only if $Q_{\Lambda }\left(\xi \right)\leq 1$ for all ξ∈R, and $E_{\Lambda }$ is an orthogonal basis of $L^{2}\left(\mu \right)$ if and only if $Q_{\Lambda }\left(\xi \right)=1$ for all ξ∈R.*


Given a finite set D⊂R, we call
$$m_{D}\left(\xi \right)=\frac{1}{♯D}\sum_{d\in D}\exp\left(2\pi i\xi d\right),\;\;\xi \in R$$
the mask of *D*. It is clear that it is just the Fourier transformation of the uniform probability measure on *D*.

**Definition** **1.**
*For two finite subsets D and S of R with the same cardinality m, we say (D,S) is a compatible pair if*

$${\left[\frac{1}{\sqrt{m}}\exp\left(2\pi ids\right)\right]}_{d\in D,s\in S}$$

*is a unitary matrix.*


The following conclusion is well known.

**Lemma** **1.**
*For two finite subsets D and S of R with the same cardinality m, the following statements are equivalent:*


(i).
*(D,S) is a compatible pair;*
(ii).
*$m_{D}\left(s_{1}-s_{2}\right)=0$ for any $s_{1}\neq s_{2}\in S;$*
(iii).
*$\sum_{s\in S}{\left|m_{D}\left(\xi +s\right)\right|}^{2}=1$ for any ξ∈R.*


In other words, (D,S) is a compatible pair if and only if *S* is a spectrum of the uniform probability measure on *D*.

Let *N* be an integer with |N|>1 and $D={\left\{d_{j}\right\}}_{j=1}^{q}$ a finite subset of Z with 0∈D. We denote by $\mu_{N,D}$ the unique invariant measure with respect to the IFS $\left\{\phi_{j}\left(x\right)=\frac{1}{N}\left(x+d_{j}\right):1\leq j\leq q\right\}$ with equal probability weights, i.e.,
$$\mu_{N,D}=\frac{1}{q}\sum_{j=1}^{q}\mu_{N,D}\circ \phi_{j}^{-1}.$$
In the sequel, we write $\mu =\mu_{N,D}$ for simplicity. Thus, we have
$$\hat{\mu }\left(\xi \right)=\prod_{j=1}^{\infty }m_{D}\left(N^{-j}\xi \right),\;\xi \in R.$$
For k≥1, we write
(1)$${\hat{\mu }}_{k}\left(\xi \right)=\prod_{j=1}^{k}m_{D}\left(N^{-j}\xi \right),\;\xi \in R.$$
Write $Y\left(m_{D}\right)=\left\{\xi \in R:m_{D}\left(\xi \right)=1\right\}$. When gcd(D)=1, we have
(2)$$Y\left(m_{D}\right)=\left\{\xi \in R:\right|m_{D}\left(\xi \right)\left|=1\right\}=Z.$$

Now, we introduce the tree structure. First, we recall some basic notation of symbolic space. Given a positive integer q>1, write $\Sigma_{q}=\left\{0,1,\cdots ,q-1\right\}$. Let $\Sigma^{*}={⋃}_{n=0}^{\infty }\Sigma_{q}^{n}$ stand for the set of all finite words, where $\Sigma_{q}^{0}=\left\{\vartheta \right\}$ denotes the set of empty words. The length of a finite word σ is the number of symbols it contains and is denoted by |σ|. The concatenation of two finite words σ and $\sigma^{\prime }$ is written as $\sigma \sigma^{\prime }$. We say σ is a prefix of $\sigma \sigma^{\prime }$. Given $\sigma =\sigma_{1}\sigma_{2}\cdots \sigma_{n}\in \Sigma^{*}$ and 1≤k≤n, write ${\sigma |}_{k}=\sigma_{1}\cdots \sigma_{k}$. The following definition will bring convenience to us.

**Definition** **2.**
*A sequence of finite words ${\left\{I_{n}\right\}}_{n\geq 1}\subset \Sigma^{*}$ is called increasing if for any n≥1, $I_{n}$ is a prefix of $I_{n+1}$ and $|I_{n+1}\left|=\right|I_{n}|+1$.*


Let C be a mapping from $\Sigma^{*}$ to Z satisfying C(ϑ)=0 and C(I)=0 if *I* ends with the symbol 0. It induces a family of mapping $F={\left\{F_{I}\right\}}_{I\in \Sigma^{*}}$ defined by
$$\begin{array}{cc} F_{I}:\;\Sigma^{*} & ⟶Z,\\ J & ⟼C\left({IJ|}_{1}\right)+NC\left({IJ|}_{2}\right)+\cdots +N^{|J|-1}C\left(IJ\right), \end{array}$$
where ${IJ|}_{i}$ is the concatenation of *I* and J|i for 1≤i≤|J|. We write $F\left(J\right)=F_{\vartheta }\left(J\right)$ for convenience. By a simple deduction, we have the following consistency: for any $I,J,K\in \Sigma^{*}$,
$$\begin{array}{cc} \; & F_{I}\left(J\right)+N^{\left|J\right|}F_{IJ}\left(K\right)\\ = & C\left({IJ|}_{1}\right)+\cdots +N^{|J|-1}C\left(IJ\right)+N^{\left|J\right|}C\left({IJK|}_{1}\right)+\cdots +N^{|JK|-1}C\left(IJK\right)\\ = & F_{I}\left(JK\right). \end{array}$$

**Definition** **3.**
*We say a countable set Λ⊂R has a (C,F) tree structure if there exists a mapping C and an associated family of mappings F defined in the above paragraph such that*

$$\Lambda =\underset{I\in \Sigma^{*}}{⋃}\left\{F\left(I\right)\right\}.$$



For $I\in \Sigma^{*}$, let $S_{I}=\left\{C\left(Ii\right):i\in \Sigma_{q}\right\}$. According to the definition of the mapping C, we have $C\left(I0\right)=0\in S_{I}$.

**Remark** **1.**
*Given a sequence of finite sets $S={\left\{S_{n}\right\}}_{n\geq 1}$, if $S_{I}=S_{n+1}$ for any $I\in \Sigma^{n}\left(n\geq 0\right)$, we obtain*

$$\Lambda =S_{1}+NS_{2}+N^{2}S_{3}\cdots .$$

*In particular, if $S_{n}=S$ for n≥1, we obtain*

$$\Lambda =S+NS+N^{2}S\cdots ,$$

*which is just the case considered by Łaba and Wang in [3].*


In this paper, we consider a countable set Λ as a spectrum candidate satisfying the following three conditions:(C1).Λ has a (C,F) tree structure.(C2).For any $I\in \Sigma^{*}$, $\left(\frac{1}{N}D,S_{I}\right)$ is a compatible pair.(C3).The set $\tilde{S}={⋃}_{I\in \Sigma^{*}}S_{I}$ is bounded.

**Remark** **2.**
*Since we only assume that $\left(\frac{1}{N}D,S_{I}\right)$ is a compatible pair with $S_{I}=\left\{C\left(Ii\right):i\in \Sigma_{q}\right\}$, the map C may not be a maximal mapping defined in [8] (Definition 2.5) even if D={0,1,⋯,q−1}.*


Now, we exploit some basic properties of Λ satisfying the conditions (C1),(C2), and (C3). The first one is the uniqueness of the tree representation.

**Proposition** **2.**
*Let N∈Z with |N|>1 and D⊂Z with 0∈D and gcd(D)=1. Assume that a countable set Λ satisfies the conditions (C1),(C2), and (C3). Then, for any $I\in \Sigma_{q}^{*}$ and $J,\;K\in \Sigma_{q}^{n}$ with n⩾0, we have $F_{I}\left(J\right)=F_{I}\left(K\right)$ if and only if J=K.*


**Proof.** We just prove the necessity. Suppose there exist $I\in \Sigma_{q}^{*}$ and $J\neq K\in \Sigma_{q}^{n}$ with n⩾1 such that $F_{I}\left(J\right)=F_{I}\left(K\right)$. Let *l* be the smallest integer with ${J|}_{l}{\neq K|}_{l}$. From $F_{I}\left(J\right)=F_{I}\left(K\right)$, it follows that
$$N^{l-1}C\left(IJ{|}_{l}\right)+\cdots +N^{n-1}C\left(IJ\right)=N^{l-1}C\left(IK{|}_{l}\right)+\cdots +N^{n-1}C\left(IK\right),$$
which implies $C\left(IJ{|}_{l}\right)\equiv C\left(IK{|}_{l}\right)\left(\;\mathrm{mod}\;N\right)$. Noting $C\left(IJ{|}_{l}\right),\;C\left(IK{|}_{l}\right)\in S_{{IJ|}_{l-1}}$, we obtain $\left(\frac{1}{N}D,S_{{IJ|}_{l-1}}\right)$ is not a compatible pair, which is a contradiction to the condition (C2). □

**Proposition** **3.**
*Let N∈Z with |N|>1 and D⊂Z with 0∈D and gcd(D)=1. Assume that a countable set Λ satisfies the conditions (C1),(C2), and (C3). Then, E(Λ) is an orthogonal set of $L^{2}\left(\mu \right)$.*


**Proof.** Given α≠β∈Λ, there exist two finite words $I,J\in \Sigma^{*}$ such that
α=F(I),β=F(J).
If |I|≠|J|, we add symbol 0 in the end of *I* or *J* to obtain |I|=|J|. Without loss of generality, we assume that $I,J\in \Sigma_{q}^{n}$ for some integer *n*. Let *l* be the smallest positive integer satisfying ${I|}_{l}{\neq J|}_{l}$. Recall that $F\left(I{|}_{l}\right)=C\left(I{|}_{1}\right)+NC\left(I{|}_{2}\right)\cdots +N^{l-1}C\left(I{|}_{l}\right)$. Then, there exists an integer $z_{0}$ such that
$$N^{-l}\left(F\left(I{|}_{l}\right)-F\left(J{|}_{l}\right)\right)=\frac{1}{N}\left(C\left(I{|}_{l}\right)-C\left(J{|}_{l}\right)\right)+z_{0}.$$
By virtue of the condition (C2), we know that $\left(\frac{1}{N}D,S_{{I|}_{l-1}}\right)$ is a compatible pair. Noting that both $C(I{|}_{l})$ and $C(J{|}_{l})$ belong to $S_{{I|}_{l-1}}$, we obtain
$$m_{D}\left(N^{-l}\left(F\left(I{|}_{l}\right)-F\left(J{|}_{l}\right)\right)\right)=m_{D}(\frac{1}{N}\left(C\left(I{|}_{l}\right)-C\left(J{|}_{l}\right)+z_{0}\right)=m_{D}\left(\frac{1}{N}\left(C\left(I{|}_{l}\right)-C\left(J{|}_{l}\right)\right)\right)=0.$$
This leads to
$$\begin{array}{cc} & \hat{\mu }\left(\alpha -\beta \right)=\hat{\mu }\left(F\left(I\right)-F\left(J\right)\right)\\ = & \prod_{j=1}^{l-1}m_{D}\left(N^{-j}\left(F\left(I\right)-F\left(J\right)\right)\right)m_{D}\left(N^{-l}\left(F\left(I{|}_{l}\right)-F\left(J{|}_{l}\right)\right)\right)\prod_{j=l+1}^{\infty }\left(N^{-j}\left(F\left(I\right)-F\left(J\right)\right)\right)\\ = & 0. \end{array}$$□

For any $I\in \Sigma_{q}^{*}$ and k⩾1, define
$$\Lambda_{I}=\left\{F_{I}\left(J\right):J\in \Sigma^{*}\right\}\;\mathrm{and}\;\;\Lambda_{I}^{k}:=\left\{F_{I}\left(J\right):\;J\in \Sigma_{q}^{k}\right\}.$$
We write $\Lambda^{k}:=\Lambda_{\vartheta }^{k}$ for simplicity. It is clear that
$$\Lambda_{I}^{k}⊊\Lambda_{I}^{k+1}.$$

From the condition (C2) and Lemma 1(ii), it follows that $E(\Lambda_{I}^{k})$ is an orthogonal set of $L^{2}\left(\mu_{k}\right)$. By (Equation 2), we obtain $\#\Lambda_{I}^{k}=q^{k}$. Noting the fact that $\dim\left(L^{2}\left(\mu_{k}\right)\right)=q^{k},$ we conclude that $E(\Lambda_{I}^{k})$ is an orthogonal basis of $L^{2}\left(\mu_{k}\right)$. In other words, $\Lambda_{I}^{k}$ is a spectrum of $\mu_{k}$. By Lemma Equation 1, we have
(3)$$\sum_{\lambda \in \Lambda_{I}^{k}}\prod_{j=1}^{k}{\left|m_{D}\left(N^{-j}\left(\xi +\lambda \right)\right)\right|}^{2}=\sum_{\lambda \in \Lambda_{I}^{k}}{\left|{\hat{\mu }}_{k}\left(\xi +\lambda \right)\right|}^{2}\equiv 1,\;\forall \;\xi \in R.$$

In fact, we have the following conclusion.

**Proposition** **4.**
*Let N∈Z with |N|>1 and D⊂Z with 0∈D and gcd(D)=1. Assume that a countable set Λ satisfies the conditions (C1),(C2), and (C3). Then, $Q_{\Lambda }\left(\xi \right)\equiv 1$ if and only if $Q_{\Lambda_{I}}\left(\xi \right)\equiv 1$ for any $I\in \Sigma^{*}$,*


**Proof.** By virtue of $\Lambda_{\vartheta }=\Lambda$, the sufficiency is obvious.Next, we prove the necessity. Given n≥1 and $I\in \Sigma_{q}^{n}$, write $B_{I}=\left\{\xi +F\left(I\right):\xi \in \left[0,1\right]\right\}$ and ${\tilde{B}}_{I}=\left\{N^{-n}\left(\xi +F\left(I\right)\right):\xi \in \left[0,1\right]\right\}$. It is easy to see that both $B_{I}$ and ${\tilde{B}}_{I}$ are compact sets. Noting the fact that ${\hat{\mu }}_{n}$ can be extended to be an entire function on the complex plane, ${\hat{\mu }}_{n}$ has at most finitely many zero points in $B_{I}$. On the other hand, recall that
$$\Lambda =\underset{I\in \Sigma_{q}^{n}}{⋃}\underset{J\in \Sigma^{*}}{⋃}\left(F\left(I\right)+N^{n}F_{I}\left(J\right)\right),\;n\geq 1.$$
Noting the fact that every integer is a period of $m_{D}$, we have ${\hat{\mu }}_{n}\left(\xi +F\left(IJ\right)\right)={\hat{\mu }}_{n}\left(\xi +F\left(I\right)\right)$ for any $I\in \Sigma_{q}^{n}$ and $J\in \Sigma_{q}^{*}$. Hence,
(4)$$\begin{array}{cc} Q_{\Lambda }\left(\xi \right)= & \sum_{\lambda \in \Lambda }{\left|{\hat{\mu }}_{n}\left(\xi +\lambda \right)\right|}^{2}{\left|\hat{\mu }\left(N^{-n}\left(\xi +\lambda \right)\right)\right|}^{2}\\ = & \sum_{I\in \Sigma_{q}^{n}}\sum_{J\in \Sigma^{*}}{\left|{\hat{\mu }}_{n}\left(\xi +F\left(I\right)\right)\right|}^{2}{\left|\hat{\mu }\left(N^{-n}\left(\xi +F\left(I\right)+N^{n}F_{I}\left(J\right)\right)\right)\right|}^{2}\\ = & \sum_{I\in \Sigma_{q}^{n}}{\left|{\hat{\mu }}_{n}\left(\xi +F\left(I\right)\right)\right|}^{2}\sum_{J\in \Sigma^{*}}{\left|\hat{\mu }\left(N^{-n}\left(\xi +F\left(I\right)\right)+F_{I}\left(J\right)\right)\right|}^{2}\\ = & \sum_{I\in \Sigma_{q}^{n}}{\left|{\hat{\mu }}_{n}\left(\xi +F\left(I\right)\right)\right|}^{2}Q_{\Lambda_{I}}\left(N^{-n}\left(\xi +F\left(I\right)\right)\right).\;\;\; \end{array}$$
In combination with (Equation 3), this means $Q_{\Lambda_{I}}\left(\xi \right)$ takes 1 on except at most finitely many points in ${\tilde{B}}_{I}$, which implies $Q_{\Lambda_{I}}\left(\xi \right)\equiv 1$ by using the continuity of $Q_{\Lambda_{I}}\left(\xi \right)$. □

In the end of this section, we define the dual IFS $\left\{\Phi_{s}\left(x\right)=\frac{1}{N}\left(x+s\right):s\in \tilde{S}\right\}$, which plays an important role in what follows. Let *T* be the invariant set of the IFS, i.e.,
$$T=\underset{s\in \tilde{S}}{⋃}\Phi_{s}\left(T\right).$$
Define $Z\left(\hat{\mu },T\right)=Z\left(\hat{\mu }\right)\cap T$, which stands for the zero point set of $\hat{\mu }$ on *T*. It is clear that $p:=\#Z(\hat{\mu },T)$ is finite.

## 3. Main Theorem

In this section, we will give our main results involving three equivalent statements. To prove the most difficult part of the proof, we prepared several lemmas including a new criterion for a spectrum candidate with a tree structure to be a spectrum of a self-similar measure. At the end of this section, we show that the new criterion is just a sufficient and necessary condition, which is stated as a corollary .

**Theorem** **2.**
*Let N∈Z with |N|>1 and D⊂Z with 0∈D and gcd(D)=1. Assume that a countable set Λ satisfies the conditions (C1),(C2), and (C3). Then, the following statements are equivalent:*


(i).
*(μ,Λ) is not a spectral pair.*
(ii).
*There exists a finite word $I\in \Sigma^{*}$ such that $\inf_{\xi \in T}Q_{\Lambda_{I}}\left(\xi \right)=0$.*
(iii).
*There exist a finite word $J\in \Sigma^{*}$, a sequence of nonzero integers ${\left\{\beta_{l}\right\}}_{l\geq 1}\subset Z,\backslash \left\{0\right\}$ and a sequence of increasing finite words ${\left\{Ji_{1}\cdots i_{l}\right\}}_{l\geq 1}\subset \Sigma^{*}$, which has a prefix J such that, for any l≥1, we have $\beta_{l+1}=\frac{1}{N}\left(\beta_{l}+C\left(Ji_{1}\cdots i_{l}\right)\right)$.*


We shall divide the proof into three parts (iii)⇒(i), (i)⇒(ii), and (ii)⇒(iii).

First, we prove (iii)⇒(i), which plays a key role in the proof of (i)⇒(ii).

**Proof of Theorem 2** (iii)⇒(i)). We shall prove $Q_{\Lambda_{J}}\left(\beta_{1}\right)=0.$ Thus, from Proposition 4, the conclusion follows.Given $\lambda \in \Lambda_{I},$ there exists a positive integer m⩾1 and $L\in \Sigma_{q}^{m}$ such that
$$\lambda =F_{I}\left(L\right)\in \Lambda_{I}^{m}.$$
Since the sequence ${\left\{\beta_{l}\right\}}_{l\geq 1}$ is nonzero, the sequence of integers ${\left\{C\left(Ji_{1}\cdots i_{l}\right)\right\}}_{l⩾1}$ has infinitely many nonzero terms. Thus, there exist infinitely many terms *l* with $i_{l}\neq 0$. Take an integer r>m with $i_{r}\neq 0$. Write $\lambda^{*}:=F_{J}\left(K\right)\in \Lambda_{J}^{r}.$ According to Proposition 2 and $i_{r}\neq 0$, we have $\lambda \neq \lambda^{*}$ and $\lambda \in \Lambda_{J}^{m}\subset \Lambda_{J}^{r}.$ From $\beta_{k+1}=N^{-1}\left(\beta_{k}+C\left(Ji_{1}\cdots i_{k}\right)\right)\left(k⩾1\right),$ it follows that
$$\begin{array}{cc} \beta_{1}+\lambda^{*}= & N(\beta_{2}+C\left({JK|}_{2}\right)+NC\left({JK|}_{3}\right)+\cdots +N^{r-2}C\left(JK\right))\\ = & \cdots \\ = & N^{r}\beta_{r+1}\in N^{r}Z, \end{array}$$
which implies ${\left|\hat{\mu_{r}}\left(\beta_{1}+\lambda^{*}\right)\right|}^{2}=1.$ Noting () and $\lambda \neq \lambda^{*}$, we have
$$\begin{array}{cc} 1⩽ & {\left|\hat{\mu_{r}}\left(\beta_{1}+\lambda^{*}\right)\right|}^{2}+{\left|\hat{\mu_{r}}\left(\beta_{1}+\lambda \right)\right|}^{2}⩽\sum_{\gamma \in \Lambda_{J}^{r}}{\left|\hat{\mu_{r}}\left(\beta_{1}+\gamma \right)\right|}^{2}=1, \end{array}$$
Thus, we obtain $|\hat{\mu_{r}}\left(\beta_{1}+\lambda \right)|=0.$ Hence,
$$\left|\hat{\mu }\left(\beta_{1}+\lambda \right)\right|=0,\;\forall \;\lambda \in \Lambda_{J}.$$
It follows that $Q_{\Lambda_{J}}\left(\beta_{1}\right)=\sum_{\lambda \in \Lambda_{J}}{\left|\hat{\mu }\left(\beta_{1}+\lambda \right)\right|}^{2}=0$. □

The following three lemmas play key roles in the proof of Theorem Equation 2
(i)⇒(ii). First, we show a new criterion for Λ to be a spectrum of μ.

**Lemma** **2.**
*Let N∈Z with |N|>1 and D⊂Z with 0∈D and gcd(D)=1. Assume that a countable set Λ satisfies the conditions (C1),(C2), and (C3). If there exists a positive number c>0 such that, for any ξ and $I\in \Sigma^{*}$, there is $\lambda_{\xi ,I}\in \Lambda_{I}$ satisfying*

$$|\hat{\mu }\left(N^{-|I|}\left(\xi +F\left(I\right)\right)+\lambda_{\xi ,I}\right)|\geq c,$$

*then (μ,Λ) is a spectral pair.*


**Proof.** Suppose (μ,Λ) is not a spectral pair. Then, there exists $\xi_{0}\in T$ such that $Q_{\Lambda }\left(\xi_{0}\right)<1.$Recall that $\Lambda^{n}=\left\{C\left(J{|}_{1}\right)+NC\left(J{|}_{2}\right)+\cdots +N^{n-1}C\left(J\right):J\in \Sigma_{q}^{n}\right\}$ for n≥1. We write $Q_{n}\left(\xi_{0}\right):=\sum_{\lambda \in \Lambda^{n}}{\left|\hat{\mu }\left(\xi_{0}+\lambda \right)\right|}^{2}$. By virtue of $\lim_{n\to \infty }\Lambda^{n}=\Lambda$ and $\Lambda_{n}\subset \Lambda_{n+1}$ for n≥1, we obtain
$$\lim_{n\to \infty }Q_{n}\left(\xi_{0}\right)=Q_{\Lambda }\left(\xi_{0}\right)\;\mathrm{and}\;Q_{n}\left(\xi_{0}\right)⩽Q_{n+1}\left(\xi_{0}\right).$$
Given a positive number ε with $\varepsilon <\frac{1}{2}\left(1-Q_{\Lambda }\left(\xi_{0}\right)\right)$, there exists an integer M⩾1 such that
(5)$$Q_{\Lambda }\left(\xi_{0}\right)-\varepsilon ⩽Q_{M}\left(\xi_{0}\right)⩽Q_{n}\left(\xi_{0}\right)⩽Q_{\Lambda }\left(\xi_{0}\right)<1,\;\forall \;n⩾M.$$
By (Equation 1) we have
$$\lim_{m\to \infty }{\hat{\mu }}_{m}\left(\xi_{0}+\lambda \right)=\hat{\mu }\left(\xi_{0}+\lambda \right),\;\;\forall \;\lambda \in \Lambda .$$
In combination with (Equation 5), we have a positive integer K⩾M+1 such that
$$\sum_{\lambda \in \Lambda^{M}}{\left|{\hat{\mu }}_{K}\left(\xi_{0}+\lambda \right)\right|}^{2}⩽\sum_{\lambda \in \Lambda^{M}}{\left|\hat{\mu }\left(\xi_{0}+\lambda \right)\right|}^{2}+\varepsilon ⩽Q_{\Lambda }\left(\xi_{0}\right)+\varepsilon .$$
According to (Equation 3), we have $\sum_{\lambda \in \Lambda^{K}}{\left|{\hat{\mu }}_{K}\left(\xi_{0}+\lambda \right)\right|}^{2}=1$. Thus,
(6)$$\begin{array}{cc} \sum_{I\in \Sigma_{q}^{K}\backslash \Sigma_{q}^{M}}{\left|{\hat{\mu }}_{K}\left(\xi_{0}+F\left(I\right)\right)\right|}^{2}\; & =\sum_{\lambda \in \Lambda^{K}}{\left|{\hat{\mu }}_{K}\left(\xi_{0}+\lambda \right)\right|}^{2}-\sum_{\lambda \in \Lambda^{M}}{\left|{\hat{\mu }}_{K}\left(\xi_{0}+\lambda \right)\right|}^{2}\\ \; & ⩾1-Q_{\Lambda }\left(\xi_{0}\right)-\varepsilon >0. \end{array}$$For any $I\in \Sigma_{q}^{K}\backslash \Sigma_{q}^{M},$ there exists $\lambda_{\xi_{0},I}\in \Lambda_{I}$ such that
(7)$$|\hat{\mu }\left(N^{-K}\left(\xi_{0}+F\left(I\right)\right)+\lambda_{\xi_{0},I}\right)|>c.$$
Write $\tilde{\Lambda }=\left\{F\left(I\right)+N^{K}\lambda_{\xi_{0},I}:\;I\in \Sigma_{q}^{K}\backslash \Sigma_{q}^{M},\;\lambda_{\xi_{0},I}\in \Lambda_{I}\right\}$. It is clear that $\tilde{\Lambda }\subset \Lambda$. Since $\left(\frac{1}{N}D,S_{I}\right)$ is a compatible pair for any $I\in \Sigma^{*}$, C(I)=0 if and only if the finite word *I* ends with the symbol 0. Then, we have
$$\Lambda^{M}\cap \tilde{\Lambda }=\emptyset .$$
In combination with (Equation 5)–(Equation 7), we obtain
$$\begin{array}{cc} \; & Q_{\Lambda }\left(\xi_{0}\right)=\sum_{\lambda \in \Lambda }{\left|\hat{\mu }\left(\xi_{0}+\lambda \right)\right|}^{2}\\ \; & \geq \sum_{\lambda \in \Lambda^{M}}{\left|\hat{\mu }\left(\xi_{0}+\lambda \right)\right|}^{2}+\sum_{\lambda \in \tilde{\Lambda }}{\left|\hat{\mu }\left(\xi_{0}+\lambda \right)\right|}^{2}\\ \; & =\sum_{\lambda \in \Lambda^{M}}{\left|\hat{\mu }\left(\xi_{0}+\lambda \right)\right|}^{2}+\sum_{I\in \Sigma^{K}\backslash \Sigma^{M}}{\left|\hat{\mu }\left(\xi_{0}+F\left(I\right)+N^{K}\lambda_{\xi_{0},I}\right)\right|}^{2}\\ & =\sum_{\lambda \in \Lambda^{M}}{\left|\hat{\mu }\left(\xi_{0}+\lambda \right)\right|}^{2}+\sum_{I\in \Sigma^{K}\backslash \Sigma^{M}}{\left|{\hat{\mu }}_{K}\left(\xi_{0}+F\left(I\right)\right)\right|}^{2}|\hat{\mu }{\left(N^{-K}\left(\xi_{0}+F\left(I\right)\right)+\lambda_{\xi_{0},I}\right|}^{2}\\ & ⩾Q_{\Lambda }\left(\xi_{0}\right)-\varepsilon +c^{2}\sum_{I\in \Sigma^{K}\backslash \Sigma^{M}}{\left|{\hat{\mu }}_{K}\left(\xi_{0}+F\left(I\right)\right)\right|}^{2}\\ & ⩾Q_{\Lambda }\left(\xi_{0}\right)-\varepsilon +c^{2}\left(1-Q_{\Lambda }\left(\xi_{0}\right)-\varepsilon \right). \end{array}$$
Letting ε→0, we obtain
$$\begin{array}{c} 0⩾c^{2}\left(1-Q_{\Lambda }\left(\xi_{0}\right)\right), \end{array}$$
which is a contradiction to $Q_{\Lambda }\left(\xi_{0}\right))<1$. □

To use Lemma Equation 2, we need the following lemma, which implies that, under some conditions for any point in *T*, there exists a path that escapes from $Z(\hat{\mu },T).$

**Lemma** **3.**
*Let N∈Z with |N|>1 and D⊂Z with 0∈D and gcd(D)=1. Assume that a countable set Λ satisfies the conditions (C1),(C2), and (C3) and $\inf_{\xi \in T}Q_{\Lambda_{I}}\left(\xi \right)>0$ for any $I\in \Sigma_{q}^{*}$. If $Z(\hat{\mu },T)\neq \emptyset$ and for any $\alpha \in Z(\hat{\mu },T)$ and $I\in \Sigma^{*}$, there exists no $K\in \Sigma^{*}$ with $\alpha +F_{I}\left(K\right)=0$, then for any ξ∈T, there exist two nonnegative integers w and v with 1⩽v⩽p+1 and a finite word $J=j_{1}\cdots j_{w+v}\in \Sigma_{q}^{*}$ satisfying the following property:*

*If w=0, we have*

$$0<|m_{D}\left(N^{-l}\left(\xi +F_{I}\left(J{|}_{l}\right)\right)\right)|<1,\;1⩽l⩽v,$$

*and $|\hat{\mu }\left(N^{-v}\left(\xi +F_{I}\left(J\right)\right)\right)|>0;$*

*If w>0, we have*

$$\begin{array}{cc} \; & m_{D}\left(N^{-l}\left(\xi +F_{I}\left(J{|}_{l}\right)\right)\right)=1,\;\;1⩽l⩽w,\\ \; & 0<|m_{D}\left(N^{-l}\left(\xi +F_{I}\left(J{|}_{l}\right)\right)\right)|<1,\;\;w+1⩽l⩽w+v, \end{array}$$

*and $|\hat{\mu }\left(N^{-w-v}\left(\xi +F_{I}\left(J\right)\right)\right)|>0$.*


**Proof.** First, we shall prove the existence of *w*. If T∩Z=∅, we take w=0. If T∩Z≠∅, since $\left(\frac{1}{N}D,S_{I}\right)$ is a compatible pair, by Lemma Equation 1(iii), there exists $j_{1}\in \Sigma_{q}$ such that
(8)$$|m_{D}\left(N^{-1}\left(\xi +F_{I}\left(j_{1}\right)\right)\right)|>0.$$
If $|m_{D}\left(N^{-1}\left(\xi +F_{I}\left(j_{1}\right)\right)\right)|<1,$ we take w=0. If $|m_{D}\left(N^{-1}\left(\xi +F_{I}\left(j_{1}\right)\right)\right)|=1$, also by Lemma Equation 1(iii), there exists $j_{2}\in \Sigma_{q}$ such that
$$|m_{D}\left(N^{-2}\left(\xi +F_{I}\left(j_{1}j_{2}\right)\right)\right)|>0.$$
If $|m_{D}\left(N^{-2}\left(\xi +F_{I}\left(j_{1}j_{2}\right)\right)\right)|<1,$ we take w=1. When $|m_{D}\left(N^{-2}\left(\xi +F_{I}\left(j_{1}j_{2}\right)\right)\right)|=1$, the process goes on. Under the process, we claim that there exists a finite sequence of symbols ${\left\{j_{n}\right\}}_{n=1}^{w}\subset \Sigma_{q}$ such that
$$m_{D}\left(N^{-l}\left(\xi +F_{I}\left(j_{1}\cdots j_{l}\right)\right)\right)=1,\;\forall \;1⩽l⩽w,$$
and
(9)$$0<\left|m_{D}\left(N^{-w-1}\left(\xi +F_{I}\left(j_{1}\cdots j_{w+1}\right)\right)\right)\right|<1,\;\;\forall \;j_{w+1}\in \Sigma_{q}.$$
Otherwise, there exists an infinite sequence ${\left\{j_{l}\right\}}_{l⩾1}\subset \Sigma_{q}$ such that $m_{D}\left(N^{-l}\left(\xi +F_{I}\left(j_{1}\cdots j_{l}\right)\right)\right)$=1 for l⩾1. By (Equation 2) and the hypothesis of the lemma, we have $N^{-l}\left(\xi +F_{I}\left(j_{1}\cdots j_{l}\right)\right)\in Z\backslash \left\{0\right\}.$ According to the proof of Theorem (Equation 2)(iii)⇒(i), we obtain $Q_{\Lambda_{I}}\left(\xi \right)=0$, which is a contradiction to the condition $\inf_{\xi \in T}Q_{\Lambda_{I}}\left(\xi \right)>0$ for any $I\in \Lambda^{*}$.Next, we shall prove the existence of *v*. We write $\tilde{J}:=j_{1}\cdots j_{w}$ and $\eta :=N^{-w}\left(\xi +F_{I}\left(\tilde{J}\right)\right)$, where $\tilde{J}=\vartheta$, $F_{I}\left(\tilde{J}\right)=0$ and η=ξ when w=0. In what follows, we define a sequence of sets ${\left\{Y_{n}\right\}}_{n\geq 0}$ by induction on *n*. Define $Y_{0}=\left\{\vartheta \right\}$, and
$$Y_{n}:=\left\{L\in \Sigma_{q}^{n}:L{|}_{n-1}\in Y_{n-1},0<\left|m_{D}\left(N^{-n}\left(\eta +F_{I\tilde{J}}\left(L\right)\right)\right)\right|<1\right\},\;n\geq 1.$$
We have the following claim. □

**Claim:** For n≥1, we have $\#Y_{n}⩾2^{n}$.

**Proof.** When n=1, since $\left(\frac{1}{N}D,S_{I\tilde{J}}\right)$ is a compatible pair, there exist two symbols $l_{1}\neq l_{2}\in \Sigma_{q}$ such that
$$0<|m_{D}\left(N^{-1}\left(\eta +F_{I\tilde{J}}\left(l_{k}\right)\right)\right)|<1,\;1⩽k⩽2.$$Thus, we obtain $\#Y_{1}⩾2$. Suppose the inequality $\#Y_{n}⩾2^{n}$ holds as n=k. Let n=k+1. For any $L\in Y_{k}$, it is clear ${L|}_{1}\in Y_{1}$. By (Equation 9), we obtain $N^{-1}\left(\eta +F_{I\tilde{J}}\left(L{|}_{1}\right)\right)\notin Z$. Thus, $N^{-k}\left(\eta +F_{I\tilde{J}}\left(L\right)\right)\notin Z$. Since $\left(\frac{1}{N}D,S_{I\tilde{J}L}\right)$ is a compatible pair, there exist at least two symbols $l_{1}\neq l_{2}\in \Sigma_{q}$ such that
$$0<|m_{D}\left(N^{-n-1}\left(\eta +F_{I\tilde{J}}\left(Ll_{k}\right)\right)\right)|<1,\;1⩽k⩽2.$$
By the arbitrariness of $L\in Y_{k}$, we obtain $\#Y_{n+1}⩾2^{n+1}$. Hence, the claim follows by induction.Together with Proposition 2, the above claim implies
$$\#\{N^{-p-1}\left(\alpha +F_{I\tilde{J}}\left(L\right):L\in Y_{p+1}\right\}=\#Y_{p+1}⩾2^{p+1}>p.$$
Thus, by $p=♯Z(\hat{\mu },T)$, there exists a finite word $L\in Y_{p+1}$ such that
$$|\hat{\mu }\left(N^{-p-1}\left(\eta +F_{I\tilde{J}}\left(L\right)\right)\right)|>0.$$
Let v⩾1 be the smallest positive integer such that $|\hat{\mu }\left(N^{-v}\left(\eta +F_{I\tilde{J}}\left(L\right)\right)\right)|>0$ for some $L=l_{1}\cdots l_{v}$. By taking $J=\tilde{J}l_{1}\cdots l_{v}$, we finish the proof. □

**Lemma** **4.**
*If T∩Z≠∅, then there exists $\alpha_{1}>0$ such that, for any integer sequence ${\left\{\theta_{i}\right\}}_{i⩾1}\subset T\cap Z$, we have*

$$\prod_{i=1}^{\infty }\left|m_{D}\left(x_{i}\right)\right|⩾\alpha_{1},$$

*where $x_{i}\in B\left(\theta_{i},N^{-i}\right)$.*


**Proof.** For any θ∈T∩Z, we have $m_{D}\left(\theta \right)=1$. On the other hand, the mask function $m_{D}$ can be extended to an entire function on the complex plane. Thus, $m_{D}$ is uniformly continuous on any compact set. Hence, there exists a positive number $c_{1}$ such that
$$\left|1-m_{D}\left(x\right)\right|=\left|m_{D}\left(\theta \right)-m_{D}\left(x\right)\right|⩽c_{1}\left|x-\theta \right|,\;\forall \;x\in \left\{\xi +y:\;\xi \in T,|y|\leq 1\right\}.$$Given a sequence ${\left\{\theta_{i}\right\}}_{i⩾1}\subset T\cap Z$, we have
$$\left|m_{D}\left(x_{i}\right)\right|⩾1-c_{1}\left|x_{i}-\theta_{i}\right|⩾1-N^{-i}c_{1},\;\forall \;x_{i}\in B\left(\theta_{i},N^{-i}\right),\;i⩾1.$$
It is clear that there exists a positive integer K>0 such that, for k⩾K, we have $N^{-k}c_{1}<\frac{1}{2}.$ Note an elementary inequality:
$$1-x⩾e^{-2x},\;0\leq x⩽\frac{1}{2}.$$
Then, we have
(10)$$\begin{array}{cc} \prod_{i=1}^{\infty }\left|m_{D}\left(x_{i}\right)\right|= & \prod_{i=1}^{K}\left|m_{D}\left(x_{i}\right)\right|\prod_{i=K+1}^{\infty }\left|m_{D}\left(x_{i}\right)\right|\\ ⩾ & {\left(\frac{1}{2}\right)}^{K}\prod_{i=K+1}^{\infty }e^{-2c_{1}N^{-i}}\\ = & {\left(\frac{1}{2}\right)}^{K}e^{\sum_{i=K+1}^{\infty }-2c_{1}N^{-i}}\\ = & {\left(\frac{1}{2}\right)}^{K}e^{-2c_{1}\frac{1}{N^{K}\left(N-1\right)}}=:\alpha_{1}>0 \end{array}$$
for all $x_{i}\in B\left(\theta_{i},N^{-i}\right)$. The proof is complete. □

**Proof of Theorem 2** (i)⇒(ii)). We expect to obtain a contradiction after assuming
(11)$$\underset{\xi \in T}{\inf}Q_{\Lambda_{I}}\left(\xi \right)>0,\;\forall \;I\in \Sigma_{q}^{*}.$$We shall prove that there is a positive number c>0 such that, for any ξ∈T and $I\in \Sigma^{*}$, there exists $\lambda_{\xi ,I}\in \Lambda_{I}$ satisfying
$$|\hat{\mu }\left(N^{-|I|}\left(\xi +F\left(I\right)\right)+\lambda_{\xi ,I}\right)|\geq c.$$If $Z(\hat{\mu },T)=\emptyset$, then $\hat{\mu }\left(\xi \right)$ has a positive lower bound on compact set *T*. Write $c:=\inf_{\xi \in T}\left|\hat{\mu }\left(\xi \right)\right|>0$. For any ξ∈T and $I\in \Sigma_{q}^{*},$ take $\lambda_{\xi ,I}=0\in \Lambda_{I}.$ Noting $N^{-|I|}\left(\xi +F\left(I\right)\right)\in T,$ we have
$$|\hat{\mu }\left(N^{-|I|}\left(\xi +F\left(I\right)\right)\right)|⩾c.$$
From Lemma Equation 2, it follows that (μ,Λ) is a spectral pair, which is a contradiction to the hypothesis. □

Next, we focus on the case $Z(\hat{\mu },T)\neq \emptyset$. We shall deal with two cases.

**Case i**. *For any $\eta \in Z(\hat{\mu },T)$ and $I\in \Sigma^{*}$, there exists $J\in \Sigma^{*}$ such that*(12)$$\eta +F_{I}\left(J\right)=0.$$

By $|\hat{\mu }\left(0\right)|=1$, there exists a positive number δ with $0<\delta_{1}<1$ such that
(13)$$\left|\hat{\mu }\left(x\right)\right|>\frac{1}{2},\;\forall \;x\in B\left(0,\delta_{1}\right).$$
Write $\delta :=\min\{\delta_{1},\frac{d}{4}\}$, where *d* denotes the smallest distance between different points in $Z\left(\hat{\mu },T\right)\cup \left(T\cap Z\right)$, i.e., $d:=\min\{\left|x-y\right|:x\neq y\in Z\left(\hat{\mu },T\right)\cup T\cap Z\}.$

We denote the set of points that has a positive distance from the zero points of $\hat{\mu }\left(\xi \right)$ in *T* by
$$P:=T\backslash \left(\underset{\theta \in Z(\hat{\mu },T)}{⋃}B\left(\theta ,\delta \right)\right).$$
It is clear that *P* is a compact set and $\alpha_{0}:=\inf_{\xi \in P}\left|\hat{\mu }\left(\xi \right)\right|>0.$ Write $\alpha :=\min\{\frac{1}{2}\alpha_{1},\alpha_{0}\}$. Given ξ∈T and $I\in \Sigma^{*}$, define $\tilde{\xi }=N^{-|I|}\left(\xi +F\left(I\right)\right)$.

If $\tilde{\xi }\in P$, we take $\lambda_{\xi ,I}=0$. Then,
(14)$$|\hat{\mu }\left(N^{-|I|}\left(\xi +F\left(I\right)\right)+\lambda_{\xi ,I}\right)\left|=\right|\hat{\mu }\left(\tilde{\xi }\right)|\geq \alpha_{0}\geq \alpha .$$

If $\tilde{\xi }\notin P$, by the definition of *P*, there exists a unique $\theta \in Z\left(\hat{\mu },T\right)\subset T\backslash \left\{0\right\}$ such that $\tilde{\xi }\in B\left(\theta ,\delta \right).$ According to (Equation 12), there exists $J\in \Sigma^{*}$ such that
(15)$$\theta +F_{I}\left(J\right)=0.$$
Take $\lambda_{\xi ,I}=F_{I}\left(J\right)$. Then, we have
(16)$$N^{-l}\left(\tilde{\xi }+F_{I}\left(J{|}_{l}\right)\right)\in B\left(N^{-l}\left(\theta +F_{I}\left(J{|}_{l}\right)\right),N^{-l}\delta \right),\;1\leq l\leq \left|J\right|.$$
On the other hand, by (Equation 15), we have
$$N^{-l}\left(\theta +F_{I}\left(J{|}_{l}\right)\right)\in Z\cap T,\;1\leq l\leq \left|J\right|.$$
In combination with Lemma Equation 4 and (Equation 16), this leads to
(17)$$\prod_{l=1}^{\left|J\right|}\left|m_{D}\left(N^{-l}\left(\theta +F_{I}\left(J{|}_{l}\right)\right)\right)\right|>\alpha_{1}.$$
Furthermore, by (Equation 16) we have
$$N^{-|J|}\left(\tilde{\xi }+F_{I}\left(J\right)\right)\in B\left(N^{-|J|}\left(\theta +F_{I}\left(J\right)\right),N^{-|J|}\delta \right)\subset B\left(0,\delta_{1}\right).$$
Then, by (Equation 13), we have $|\hat{\mu }\left(N^{-|J|}\left(\tilde{\xi }+F_{I}\left(J\right)\right)\right)|\geq \frac{1}{2}$. Together with (Equation 17), this inequality implies
(18)$$\begin{array}{cc} |\hat{\mu }\left(N^{-|I|}\left(\xi +F\left(I\right)\right)+\lambda_{\xi ,I}\right)|= & \left|\hat{\mu }\left(\tilde{\xi }+F_{I}\left(J\right)\right)\right|\\ = & \prod_{l=1}^{\left|J\right|}\left|m_{D}\left(N^{-l}\left(\theta +F_{I}\left(J{|}_{l}\right)\right)\right)\right|\left|\hat{\mu }\left(N^{-|J|}\left(\tilde{\xi }+F_{I}\left(J\right)\right)\right)\right|\\ ⩾ & \frac{1}{2}\alpha_{1}\\ ⩾ & \alpha . \end{array}$$

**Case ii**: *There exist $\eta^{*}\in Z\left(\hat{\mu },T\right)$ and $I\in \Sigma^{*}$ such that, for any $J\in \Sigma^{*}$, we have*(19)$$\eta^{*}+F_{I}\left(J\right)\neq 0.$$
Recall that $\tilde{S}={⋃}_{I\in \Sigma^{*}}S_{I}$ and $p=♯Z(\hat{\mu },T)$. Let
$$U:=\overset{p+1}{\underset{l=1}{⋃}}\left\{N^{-l}\left(\theta +\lambda \right):\;\lambda \in \tilde{S}+N\tilde{S}+\cdots +N^{l-1}\tilde{S},\;\theta \in Z\left(\hat{\mu },T\right)\cup \left(T\cap Z\right)\right\}.$$
Furthermore, we write
$$V=\left\{x\in U:\;\left|m_{D}\left(x\right)\right|\neq 0\right\}\;\mathrm{and}\;W=\left\{x\in V:\left|\hat{\mu }\left(x\right)\right|\neq 0\right\}.$$
It is clear W⊂V⊂U⊂T. Since $\left(\frac{1}{N}D,S_{I}\right)$ is a compatible pair for any $I\in \Sigma^{*}$, we obtain V≠∅.

Next, we shall prove W≠∅.

**Claim 1:** There exists $a\in \tilde{S}$ such that
$$0<|m_{D}\left(N^{-1}\left(\eta^{*}+a\right)\right)|<1.$$

**Proof.** If T∩Z=∅, then we have $\sup\{\left|m_{D}\left(\eta \right)\right|:\eta \in T\}<1$ by noting that *T* is compact. A trivial fact that $N^{-1}\left(\eta^{*}+a\right)\in T$ for any $a\in \tilde{S}$ implies the claim is true.When T∩Z≠∅, suppose the claim is false. Since $\left(\frac{1}{N}D,S_{I}\right)$ is a compatible pair, by Lemma 1(iii) for $\eta^{*}\in Z\left(\hat{\mu },T\right)$, there exists $j_{1}\in \Sigma_{q}$ such that $m_{D}\left(N^{-1}\left(\eta^{*}+F_{I}\left(j_{1}\right)\right)\right)=1$. By (Equation 2) and (Equation 19), we obtain $N^{-1}\left(\eta^{*}+F_{I}\left(j_{1}\right)\right)\in \left(T\cap Z\right)\backslash \left\{0\right\}$. Furthermore, there exists $j_{2}\in \Sigma_{q}$ such that $m_{D}\left(N^{-2}\left(\eta^{*}+F_{I}\left(j_{1}j_{2}\right)\right)\right)=1$, which implies $N^{-2}\left(\eta^{*}+F_{I}\left(j_{1}j_{2}\right)\right)\in \left(T\cap Z\right)\backslash \left\{0\right\}$. Repeating this process, we obtain a sequence of symbols ${\left\{j_{l}\right\}}_{l⩾1}\subset \Sigma_{q}$ such that
$$N^{-l}\left(\eta^{*}+F_{I}\left(j_{1}\cdots j_{l}\right)\right)\in \left(T\cap Z\right)\backslash \left\{0\right\},\;l⩾1.$$
By a similar argument in the proof of Theorem (Equation 2)(iii)⇒(i), we obtain $Q_{\Lambda_{I}}\left(\eta^{*}\right)=0$, which implies a contradiction to (Equation 11). The claim is proven.Next, we define a sequence of set ${\left\{Y_{n}\right\}}_{n\geq 0}$ by induction on *n*. Let $Y_{0}:=\left\{\eta^{*}\right\},$ and
$$Y_{n}:=\left\{N^{-1}\left(\eta +a\right):0<\left|m_{D}\left(N^{-1}\left(\eta +a\right)\right)\right|<1,\;\eta \in Y_{n-1},\;a\in \tilde{S}\right\},\;n⩾1.$$
By a similar argument in the proof of the claim in Lemma (Equation 3), we obtain $\#Y_{n}\geq 2^{n}$ for 1≤n≤p+1. On the other hand, for any $\eta \in Y_{p+1},$ there exists $\lambda \in \tilde{S}+N\tilde{S}+\cdots +N^{p}\tilde{S}$ such that $\eta =N^{-p-1}\left(\eta^{*}+\lambda \right)$ and $0<|m_{D}\left(\eta \right)|<1$, which implies $Y_{p+1}\subset V.$ Then, we conclude
$$\#V⩾\#Y_{p+1}⩾2^{p+1}>p.$$
Recall that *p* is the number of zero points of $\hat{\mu }\left(\xi \right)$ on compact *T*. Then, we obtain W≠∅.Noting that W⊂V⊂U and *U* is a finite set, it is obvious that both *W* and *V* are finite sets. Write
$$\begin{array}{cc} \; & \alpha_{2}:=\min\left\{\left|m_{D}\left(\eta \right)\right)\right|\neq 0:\eta \in V\}>0,\\ \; & \alpha_{3}:=\min\left\{\left|\hat{\mu }\left(\eta \right)\right|\neq 0:\eta \in W\right\}>0. \end{array}$$
Then, there exists a positive number $\delta_{2}>0$ such that, for any η∈V and ω∈W, we have
(20)$$\left|m_{D}\left(x\right)\right|>\frac{1}{2}\alpha_{2},\;\forall \;x\in B\left(\eta ,\delta_{2}\right),$$
(21)$$\left|\hat{\mu }\left(x\right)\right|>\frac{1}{2}\alpha_{3},\;\forall \;x\in B\left(\omega ,\delta_{2}\right).$$
Write $\tilde{\delta }:=\min\left\{\delta_{1},\delta_{2},\frac{d}{4}\right\}$. We let $\tilde{P}:=T\backslash \left({⋃}_{\theta \in Z(\hat{\mu },T)}B\left(\theta ,\tilde{\delta }\right)\right)$ denote the set of points that has a positive distance (at least $\tilde{\delta }$) from the zero points of $\hat{\mu }\left(\xi \right)$ in *T*. It is clear that $\tilde{P}$ is a compact set and $\alpha_{4}:=\inf_{\xi \in \tilde{P}}\left|\hat{\mu }\left(\xi \right)\right|>0.$ We write
$$\tilde{\alpha }:=\min\left\{\alpha_{1}\frac{\alpha_{3}}{2}{\left(\frac{\alpha_{2}}{2}\right)}^{p+1},\alpha_{4}\right\},$$
where $\alpha_{1}$ comes from Lemma Equation 4.Given ξ∈T and $I\in \Sigma_{q}^{*}$, write $\tilde{\xi }:=N^{-|I|}\left(\xi +F\left(I\right)\right).$If $\tilde{\xi }\in \tilde{P},$ we take $\lambda_{\xi ,I}=0\in \Lambda_{I}.$ Then, we have
(22)$$\left|\hat{\mu }\left(N^{-|I|}\left(\xi +F\left(I\right)\right)+\lambda_{\xi ,I}\right)\right|=\left|\hat{\mu }\left(\tilde{\xi }\right)\right|⩾\alpha_{4}⩾\tilde{\alpha }.$$
If $\tilde{\xi }\notin \tilde{P},$ there exists $\theta \in Z(\hat{\mu },T)$ such that $\tilde{\xi }\in B\left(\theta ,\tilde{\delta }\right).$ If there exists $J\in \Sigma^{*}$ such that
$$\theta +F_{I}\left(J\right)=0,$$
we take $\lambda_{\xi ,I}=F_{I}\left(J\right).$ Then, by a similar argument as (Equation 18), we have
(23)$$|\hat{\mu }\left(N^{-|I|}\left(\xi +F\left(I\right)\right)+\lambda_{\xi ,I}\right)|\geq \alpha .$$
If there is no $J\in \Sigma^{*}$ such that
$$\theta +F_{I}\left(J\right)=0,$$
by Lemma Equation 3 there exist two integers 0⩽w<∞,1⩽v⩽p+1 and a finite word $J:=j_{1}\cdots j_{w+v}\in \Sigma_{q}^{*}$ such that when w=0, we have
(24)$$0<|m_{D}\left(N^{-l}\left(\theta +F_{I}\left(J{|}_{l}\right)\right)\right)|<1,\;1⩽l⩽v,$$
and $|\hat{\mu }\left(N^{-v}\left(\theta +F_{I}\left(J\right)\right)\right)|>0$; when w>0, we have
(25)$$\begin{array}{ccc} \; & m_{D}\left(N^{-l}\left(\theta +F_{I}\left(J{|}_{l}\right)\right)\right)=1, & 1⩽l⩽w, \end{array}$$
(26)$$\begin{array}{ccc} \; & 0<|m_{D}\left(N^{-l}\left(\theta +F_{I}\left(J{|}_{l}\right)\right)\right)|<1, & w+1⩽l⩽w+v, \end{array}$$
and $|\hat{\mu }\left(N^{-w-v}\left(\theta +F_{I}\left(J\right)\right)\right)|>0.$Take $\lambda_{\xi ,I}:=F_{I}\left(J\right)$. In the case w=0, since $\tilde{\xi }\in B\left(\theta ,\tilde{\delta }\right)$, it is obvious that
(27)$$N^{-l}\left(\tilde{\xi }+F_{I}\left(J{|}_{l}\right)\right)\in B\left(N^{-l}\left(\theta +F_{I}\left(J{|}_{l}\right)\right),N^{-l}\tilde{\delta }\right),\;1⩽l⩽v.$$
Noting that $\theta \in Z\left(\hat{\mu },T\right)\cup \left(T\cap Z\right)$, by (Equation 24) we obtain
$$N^{-l}\left(\theta +F_{I}\left(J{|}_{l}\right)\right)\in V,\;1⩽l⩽v.$$
Together with (Equation 20) and (Equation 27), the above inequality implies
(28)$$\left|m_{D}\left(N^{-l}\left(\tilde{\xi }+F_{I}\left(J{|}_{l}\right)\right)\right)\right|>\frac{\alpha_{2}}{2},\;1⩽l⩽v.$$
Furthermore, since $N^{-v}\left(\theta +F_{I}\left(J\right)\right)\in V$ and $|\hat{\mu }\left(N^{-v}\left(\theta +F_{I}\left(J\right)\right)\right)|>0,$ we have $N^{-v}\left(\theta +F_{I}\left(J\right)\right)\in W$ and $N^{-v}\left(\tilde{\xi }+F_{I}\left(J\right)\right)\in B\left(N^{-v}\left(\theta +F_{I}\left(J\right)\right),N^{-v}\tilde{\delta }\right)$. From (Equation 21) it follows that
(29)$$\left|\hat{\mu }\left(N^{-v}\left(\tilde{\xi }+F_{I}\left(J{|}_{l}\right)\right)\right)\right|⩾\frac{\alpha_{3}}{2}.$$
In combination with (Equation 28) this yields
(30)$$\begin{array}{cc} |\hat{\mu }\left(N^{-|I|}\left(\xi +F\left(I\right)\right)+\lambda_{\xi ,I}\right)|= & \left|\hat{\mu }\left(\tilde{\xi }+\lambda_{\xi ,I}\right)\right|\\ = & \prod_{i=1}^{\infty }\left|m_{D}\left(N^{-i}\left(\tilde{\xi }+F_{I}\left(J\right)\right)\right)\right|\\ = & \prod_{i=1}^{v}|m_{D}\left(N^{-i}\left(\tilde{\xi }+F_{I}\left(J\right)\right)\right|\left|\hat{\mu }\left(N^{-v}\left(\tilde{\xi }+F_{I}\left(J\right)\right)\right)\right|\\ ⩾ & \frac{\alpha_{3}}{2}{\left(\frac{\alpha_{2}}{2}\right)}^{p+1}\\ ⩾ & \tilde{\alpha }. \end{array}$$In the case w>0, we shall divide the product into three parts
(31)$$\begin{array}{cc} \; & \left|\hat{\mu }\left(N^{-|I|}\left(\xi +F\left(I\right)\right)+\lambda_{\xi ,I}\right)\right|\\ = & \prod_{i=1}^{\infty }\left|m_{D}\left(N^{-i}\left(\tilde{\xi }+F_{I}\left(J\right)\right)\right)\right|\\ = & \prod_{i=1}^{w}\left|m_{D}\left(N^{-i}\left(\tilde{\xi }+F_{I}\left(J\right)\right)\right)\right|\prod_{i=w+1}^{w+v}\left|m_{D}\left(N^{-i}\left(\tilde{\xi }+F_{I}\left(J\right)\right)\right)\right|\left|\hat{\mu }\left(N^{-w-v}\left(\tilde{\xi }+F_{I}\left(J\right)\right)\right)\right|. \end{array}$$
By (Equation 2) and (Equation 25) we have
(32)$$N^{-l}\left(\theta +F_{I}\left(J{|}_{l}\right)\right)\in T\cap Z,\;1⩽l⩽w.$$
Noting $\tilde{\xi }\in B\left(\theta ,\tilde{\delta }\right)$, we have
$$N^{-l}\left(\tilde{\xi }+F_{I}\left(J{|}_{l}\right)\right)\in B\left(N^{-l}\left(\theta +F_{I}\left(J{|}_{l}\right)\right),\;N^{-l}\tilde{\delta }\right),\;1⩽l⩽w.$$
Thus, by (Equation 10), we obtain
(33)$$\prod_{l=1}^{w}\left|m_{D}\left(N^{-l}\left(\tilde{\xi }+F_{I}\left(J{|}_{l}\right)\right)\right)\right|⩾\alpha_{1}.$$
By (Equation 32), we have $N^{-w}\left(\theta +F_{I}\left(J{|}_{w}\right)\right)\in Z\left(\hat{\mu },T\right)\cup \left(T\cap Z\right)$. Then, by (Equation 26) we have
$$N^{-l}\left(\theta +F_{I}\left(J{|}_{l}\right)\right)\in V,\;w+1⩽l⩽w+v$$
and
(34)$$N^{-l}\left(\tilde{\xi }+F_{I}\left(J{|}_{l}\right)\right)\in B\left(N^{-l}\left(\theta +F_{I}\left(J{|}_{l}\right)\right),N^{-l}\tilde{\delta }\right),\;w+1⩽l⩽w+v.$$
By (Equation 20) and (Equation 21), we obtain
(35)$$\left|m_{D}\left(N^{-l}\left(\tilde{\xi }+F_{I}\left(J{|}_{l}\right)\right)\right)\right|>\frac{\alpha_{2}}{2},\;w+1⩽l⩽w+v,$$
and
$$\left|\hat{\mu }\left(N^{-w-v}\left(\tilde{\xi }+F_{I}\left(J\right)\right)\right)\right|⩾\frac{\alpha_{3}}{2}.$$
Together with (Equation 31), (Equation 33), and (Equation 35), the above inequality yields
(36)$$\left|\hat{\mu }\left(N^{-|I|}\left(\xi +F\left(I\right)\right)+\lambda_{\xi ,I}\right)\right|\geq \alpha_{1}{\left(\frac{\alpha_{2}}{2}\right)}^{p+1}\frac{\alpha_{3}}{2}\geq \tilde{\alpha }.$$
In combination with (Equation 14), (Equation 18), (Equation 22), (Equation 23), (Equation 30), and (Equation 36), by Lemma Equation 2, we obtain (μ,Λ) is a spectral pair, which is a contradiction to our hypothesis. We finish the proof of (i)⇒(ii) in Theorem (Equation 2) □

Finally, we shall prove Theorem (Equation 2) (ii)⇒(iii).

Since *T* is compact, there exists $\xi^{*}\in T$ such that $Q_{\Lambda_{I}}\left(\xi^{*}\right)=0.$ Write
$$X:=\{\xi \in T:\hat{\mu }\left(\xi \right)=0\;\mathrm{and}\;m_{D}\left(\xi \right)\neq 0\}.$$
It is clear that 0∉X. Since $\left(\frac{1}{N}D,S_{I}\right)$ is a compatible pair, by Lemma (Equation 1), there exists an integer $j\in \Sigma_{q}$ with $m_{D}\left(\frac{1}{N}\left(\xi^{*}+F_{I}\left(j\right)\right)\right)\neq 0$. Noting that
$$0=Q_{\Lambda_{I}}\left(\xi^{*}\right)=\Sigma_{\lambda \in \Lambda_{I}}|\hat{\mu }\left(\xi^{*}+\lambda \right){|}^{2}\geq |m_{D}\left(\frac{1}{N}\left(\xi^{*}+F_{I}\left(j\right)\right)\right){|}^{2}{\left|\hat{\mu }\left(\frac{1}{N}\left(\xi^{*}+F_{I}\left(j\right)\right)\right)\right|}^{2},$$
we obtain $\hat{\mu }\left(\frac{1}{N}\left(\xi^{*}+F_{I}\left(j\right)\right)\right)=0$. By virtue of $\xi^{*}\in T$, we have $\frac{1}{N}\left(\xi^{*}+F_{I}\left(j\right)\right)\in T$. Hence, *X* is nonempty.

Next, we define a sequence of the subset of *X* by induction on *n*. Define $X_{0}:=\left\{\xi^{*}\right\}$ and
$$X_{n+1}:=\left\{N^{-n-1}\left(\xi +F_{I}\left(J\right)\right)\in X:\;N^{-n}\left(\xi +F_{I}\left(J{|}_{n}\right)\right)\in X_{n},\;J\in \Sigma_{q}^{n+1}\right\},\;n\geq 0.$$
We have the following conclusion.

**Claim** 2: $\#X_{n+1}⩾\#X_{n}$, n⩾0.

**Proof.** When n=0, by the definition of $Q_{\Lambda_{I}}\left(\xi^{*}\right)$, we have
$$\begin{array}{cc} \; & 0=Q_{\Lambda_{I}}\left(\xi^{*}\right)=\sum_{j_{1}\in \Sigma_{q}}{\left|m_{D}\left(N^{-1}\left(\xi^{*}+F_{I}\left(j_{1}\right)\right)\right)\right|}^{2}\cdot Q_{\Lambda_{Ij_{1}}}\left(N^{-1}\left(\xi^{*}+F_{I}\left(j_{1}\right)\right)\right). \end{array}$$
Noting that $\left(\frac{1}{N}D,S_{I}\right)$ is a compatible pair, Lemma (Equation 1)(iii) implies that there exists at least one symbol $j_{1}\in \Sigma_{q}$ such that $|m_{D}\left(N^{-1}\left(\xi^{*}+F_{I}\left(j_{1}\right)\right)\right)|>0,$ which implies $Q_{\Lambda_{Ij_{1}}}\left(N^{-1}\left(\xi^{*}+F_{I}\left(j_{1}\right)\right)\right)=0.$ Hence, we have $\hat{\mu }\left(N^{-1}\left(\xi^{*}+F_{I}\left(j_{1}\right)\right)\right)=0$. This leads to $\#X_{1}⩾\#X_{0}.$ Suppose Claim 2 holds for n=k−1. Then, $X_{k}$ is nonempty. For any $y\in X_{k}$, there exists $\tilde{J}\in \Sigma_{q}^{k}$ such that $y=N^{-k}\left(\xi^{*}+F_{I}\left(\tilde{J}\right)\right)$ and
$$\prod_{i=1}^{k}\left|m_{D}\left(N^{-i}\left(\xi^{*}+F_{I}\left(\tilde{J}{|}_{i}\right)\right)\right)\right|>0.$$
By (Equation 1) and (Equation 4), we have
$$\begin{array}{c} 0=Q_{\Lambda_{I}}\left(\xi^{*}\right)=\sum_{\tilde{J}\in \Sigma_{q}^{k}}\prod_{i=1}^{k}{\left|m_{D}\left(N^{-i}\left(\xi^{*}+F_{I}\left(\tilde{J}{|}_{i}\right)\right)\right)\right|}^{2}\cdot Q_{\Lambda_{I\tilde{J}}}\left(N^{-k}\left(\xi^{*}+F_{I}\left(\tilde{J}\right)\right)\right). \end{array}$$
Then, we obtain $Q_{\Lambda_{IJ}}\left(N^{-k}\left(\xi^{*}+F_{I}\left(\tilde{J}\right)\right)\right)=0.$ By a similar argument, we have
$$\begin{array}{cc} 0= & Q_{\Lambda_{I\tilde{J}}}\left(N^{-k}\left(\xi^{*}+F_{I}\left(\tilde{J}\right)\right)\right)\\ = & \sum_{j_{k+1}\in \Sigma_{q}}{\left|m_{D}\left(N^{-k-1}\left(\xi^{*}+F_{I}\left(\tilde{J}j_{k+1}\right)\right)\right)\right|}^{2}\cdot Q_{\Lambda_{I\tilde{J}j_{k+1}}}\left(N^{-k-1}\left(\xi^{*}+F_{I}\left(\tilde{J}j_{k+1}\right)\right)\right). \end{array}$$
Noting that $\left(\frac{1}{N}D,S_{I\tilde{J}}\right)$ is a compatible pair, by Lemma (Equation 1)(iii), there exists at least one symbol $j_{k+1}\in \Sigma_{q}$ such that
$$|m_{D}\left(N^{-k-1}\left(\xi^{*}+F_{I}\left(\tilde{J}j_{k+1}\right)\right)\right)|>0.$$
Hence, $Q_{\Lambda_{I\tilde{J}j_{k+1}}}\left(N^{-k-1}\left(\xi^{*}+F_{I}\left(\tilde{J}j_{k+1}\right)\right)\right)=0,$ which implies $\hat{\mu }\left(N^{-k-1}\left(\xi^{*}+F_{I}\left(\tilde{J}j_{k+1}\right)\right)\right)=0.$ Thus, we obtain
$$N^{-k-1}\left(\xi^{*}+F_{I}\left(\tilde{J}j_{k+1}\right)\right)\in X_{k+1}.$$
If we consider $N^{-n-1}\left(\xi^{*}+F_{I}\left(\tilde{J}j_{n+1}\right)\right)$ as a “next generation” of $N^{-n}\left(\xi^{*}+F_{I}\left(\tilde{J}\right)\right)$ for n≥1, Proposition 2 implies that different points of $X_{k}$ have different “next generations”. Thus, we obtain $\#X_{k+1}⩾\#X_{k}$, which implies Claim 2 is true.By noting the fact that *X* is a subset of the finite set $Z(\hat{\mu },T)$, there exists a positive integer h∈N such that
(37)$$\#X_{h+m}=\#X_{h},\;m\geq 1.$$From the above argument, it follows that for any $y=N^{-n}\left(\xi^{*}+F_{I}\left(j_{1}\cdots j_{n}\right)\right)\in X_{n}$, if there exists a symbols $j_{n+1}\in \Sigma_{q}$ such that $|m_{D}\left(N^{-n-1}\left(\xi^{*}+F_{I}\left(j_{1}\cdots j_{n}j_{n+1}\right)\right)\right)|>0$, then *y* has a “next generation” $N^{-n-1}\left(\xi^{*}+F_{I}\left(j_{1}\cdots j_{n}j_{n+1}\right)\right)\in X_{n+1}$. Noting that $\left(\frac{1}{N}D,S_{Ij_{1}\cdots j_{n}}\right)$ is a compatible pair, by Lemma Equation 1 (iii), we have
$$\Sigma_{j_{n+1}\in \Sigma_{q}}{\left|m_{D}\left(N^{-1}\left(y+C\left(IJj_{n+1}\right)\right)\right)\right|}^{2}=1.$$
In combination with (Equation 37), we conclude that for any n≥h, there exists only one symbol $j_{n+1}\in \Sigma_{q}$ such that $|m_{D}\left(N^{-1}\left(y+C\left(IJj_{n+1}\right)\right)\right|\neq 0.$ In fact, $|m_{D}\left(N^{-1}\left(y+C\left(IJj_{n+1}\right)\right)\right|=1$. Then, we obtain
$$N^{-n-1}\left(\xi^{*}+F_{I}\left(Jj_{n+1}\right)\right)=N^{-1}(y+C\left(IJj_{n+1}\right)\in Z.$$
Continuing the process, we obtain a sequence of symbols ${\left\{j_{h+l}\right\}}_{l⩾1}\subset \Sigma_{q}$, such that
$$N^{-h-l}\left(\xi^{*}+F_{I}\left(Jj_{h+1}\cdots j_{h+l}\right)\right)\in Z,\;l⩾1.$$
Define $\beta_{1}:=N^{-h}\left(\xi^{*}+F_{I}\left(J\right)\right)$ and
$$\beta_{l}:=N^{-h-l+1}\left(\xi^{*}+F_{I}\left(Jj_{h+1}\cdots j_{h+l-1}\right)\right),\;l⩾2.$$
It is clear $\beta_{l}\in X_{h+l-1}$, which implies $\beta_{l}$ is nonzero. Thus, the sequence of nonzero integers ${\left\{\beta_{l}\right\}}_{l⩾1}$ and the increasing sequence of finite words ${\left\{Jj_{h+1}\cdots j_{h+l}\right\}}_{l⩾1}$ with the prefix *J* fulfill the request. □

As a corollary of Lemma Equation 2 and Theorem Equation 2, we obtain another necessary and sufficient condition for Λ to be a spectrum of μ.

**Proposition** **5.**
*Let N∈Z with |N|>1 and D⊂Z with 0∈D and gcd(D)=1. Assume that a countable set Λ satisfies the conditions (C1),(C2), and (C3). Then, (μ,Λ) is a spectral pair if and only if there exists a positive number c>0 such that, for any ξ and $I\in \Sigma^{*}$, there is $\lambda_{\xi ,I}\in \Lambda_{I}$ satisfying*

$$|\hat{\mu }\left(N^{-|I|}\left(\xi +F\left(I\right)\right)+\lambda_{\xi ,I}\right)|\geq c.$$



**Proof.** The sufficiency follows from Lemma Equation 2. We just prove the necessity here. Suppose that (μ,Λ) is a spectral pair. By Propositions 3 and 4, we obtain, for any $I\in \Sigma^{*}$,
$$Q_{\Lambda_{I}}\left(\xi \right)\equiv 1,\;\xi \in R.$$
By a similar argument in the proof of Theorem (Equation 2) (i)⇒(ii), for any ξ∈T and I∈T, there exists $\lambda_{\xi ,I}\in \Lambda_{I}$ such that
$$|\hat{\mu }\left(N^{-|I|}\left(\xi +F\left(I\right)\right)+\lambda_{\xi ,I}\right)|\geq c.$$
We finish the proof. □

## 4. An Example

In this section, we construct a self-similar measure and a set Λ(N,B) with a tree structure. Neither the criterion of Łaba and Wang (Theorem Equation 1) nor that of Strichartz ([24]) are applicable to this set Λ(N,B). However, we show that there does not exist an infinite orbit ${\left\{\beta_{l}\right\}}_{l\geq 1}\subset Z\backslash \left\{0\right\}$ associated with the dual IFS (see Theorem Equation 3), which implies Λ(N,B) is a spectrum by Theorem Equation 2.

**Example** **1.**
*Let N=6 and D={0,1,2}. Write μ for the invariant measure associated with the IFS $\left\{\phi_{1},\phi_{2},\phi_{3}\right\}$ defined by*

$$\phi_{1}\left(x\right)=\frac{1}{6}x,\;\phi_{2}\left(x\right)=\frac{1}{6}\left(x+1\right),\;\phi_{3}\left(x\right)=\frac{1}{6}\left(x+2\right).$$

*Let $B_{1}=\left\{0,8,22\right\},B_{2}=\left\{0,22,38\right\}$, $B_{3}=\left\{0,8,52\right\}$, and $B_{4}=\left\{0,38,52\right\}$. By Lemma Equation 1, a simple induction implies that $\left(\frac{1}{6}D,B_{i}\right)$ is a compatible pair for 1≤i≤4. Noting*

$$\frac{1}{6}\left(4+8\right)=2,\;\frac{1}{6}\left(2+22\right)=4,\;\frac{1}{6}\left(4+8\right)=2,\;\frac{1}{6}\left(2+22\right)=4,\;\cdots ,$$


$$\frac{1}{6}\left(10+38\right)=8,\;\frac{1}{6}\left(8+52\right)=10,\;\frac{1}{6}\left(10+38\right)=8,\;\frac{1}{6}\left(8+52\right)=10,\;\cdots ,$$

*we see that both $\Lambda (6,B_{1})$ and $\Lambda (6,B_{4})$ have an infinite iterated nonzero integer sequence, where $\Lambda \left(N,S\right):=S+NS+N^{2}S+\cdots \mathrm{finite}\;\mathrm{sum}$. Thus, by Theorem Equation 1 or by Theorem Equation 2, we conclude that both $\Lambda (6,B_{1})$ and $\Lambda (6,B_{4})$ are not a spectrum of μ. We consider the following set defined by $\left\{B_{i}:1\leq i\leq 4\right\}$.*

(38)
$$\begin{array}{cc} \; & \Lambda \left(N,B\right):=B_{1}+\underset{B_{2}\;\mathrm{and}\;B_{3}\;\mathrm{repeat}\;1\;\mathrm{time}}{\underset{︸}{NB_{2}+N^{2}B_{3}}}+\\ & N^{3}B_{4}+\underset{B_{3}\;\mathrm{and}\;B_{2}\;\mathrm{repeat}\;2\;\mathrm{times}\;}{\underset{︸}{N^{4}B_{3}+N^{5}B_{2}+N^{6}B_{3}+N^{7}B_{2}}}+\\ & N^{8}B_{1}+\underset{B_{2}\;\mathrm{and}\;B_{3}\;\mathrm{repeat}\;2^{2}\;\mathrm{times}}{\underset{︸}{N^{9}B_{2}+N^{10}B_{3}+N^{11}B_{2}+N^{12}B_{3}+N^{13}B_{2}+N^{14}B_{3}+N^{15}B_{2}+N^{16}B_{3}}}+\\ & N^{17}B_{4}+\underset{B_{3}\;\mathrm{and}\;B_{2}\;\mathrm{repeat}\;2^{3}\;\mathrm{times}}{\underset{︸}{N^{18}B_{3}+N^{19}B_{2}+\cdots +N^{32}B_{3}+N^{33}B_{2}}}+\cdots \;\;\;\;\;(\mathrm{finite}\;\mathrm{sum}). \end{array}$$

*According to Remark Equation 1, it is clear that Theorem Equation 1 cannot work. We shall show Λ(N,B) is a spectrum of μ by Theorem Equation 2 in the following Theorem Equation 3. Then, we show that Strichartz’s criterion (Theorem 2.8 in [24]) is not appropriate by proving the following Theorem Equation 4.*


Let $A_{n}$ denote the set of coefficients of $N^{n}\left(n\geq 0\right)$ in (Equation 38). Given two integers *l* and *k* with l>k≥0, we write
(39)$$\Lambda_{k}^{l}:=A_{k}+NA_{k+1}+N^{2}A_{k+2}+\cdots +N^{l-k-1}A_{l-1}.$$
We also write $\Lambda^{k}:=\Lambda_{0}^{k}$ for simplicity. For three integers m,n, and *k* with 0≤m<n<k, we have
(40)$$\begin{array}{cc} \; & \Lambda_{m}^{n}+N^{n-m}\Lambda_{n}^{k}\\ = & A_{m}+NA_{m+1}+\cdots +N^{n-m-1}A_{n-1}+N^{n-m}A_{n}+\cdots +N^{k-m-1}A_{k-1}\\ = & \Lambda_{m}^{k}. \end{array}$$

**Theorem** **3.**
*Given nonzero integer sequence ${\left\{\beta_{i}\right\}}_{i\geq 1}$, then, for any integer M>0, there exists an integer i⩾M such that*

$$\beta_{i+1}\neq N^{-1}\left(\beta_{i}+a_{i}\right),$$

*for any $a_{i}\in A_{i}$.*


**Proof.** Suppose that there exists a positive integer *M* such that, for any i>M, we have $\beta_{i+1}=6^{-1}\left(\beta_{i}+a_{i}\right)$. Let $T_{0}$ be the self-similar set generated by the dual IFS $\left\{\frac{1}{6}\left(x+s\right):\;s\in {⋃}_{j=1}^{4}B_{j}\right\}$.According to the definition of the attractor $T_{0}$, there exists a positive integer *K* such that, for any i≥K, $\beta_{i}$ belongs to a neighborhood of $T_{0}$, i.e.,
$$\beta_{i}\in \left(-1,\frac{53}{5}\right).$$Recall a fact that ${⋃}_{i=0}^{\infty }A_{i}=\left\{0,8,22,38,52\right\}$. Then, $\beta_{K+1}=6^{-1}\left(\beta_{K}+a_{K}\right)$ with $a_{K}\in {⋃}_{j=1}^{4}B_{j}$ implies $\beta_{K}\in \left\{2,4,6,8,10\right\}$. By noting that $\beta_{K+2}=6^{-1}\left(\beta_{K+1}+a_{K+1}\right)$ with $a_{K+1}\in {⋃}_{j=1}^{4}B_{j}$ implies $\beta_{K}\neq 6$, hence $\beta_{K}\in \left\{2,4,8,10\right\}.$ If $\beta_{K}=2$, then
$$a_{K}=22,\;a_{K+1}=8,\;a_{K+2}=22,\;a_{K+3}=8,\;\cdots .$$
Hence, $\left\{8,22\right\}\cap A_{i}\neq \emptyset$ for all i≥K, which contradicts that $\left\{8,22\right\}\cap B_{4}=\emptyset$ and $B_{4}=A_{i}$ for infinitely many *i*. Hence, $\beta_{K}\in \left\{4,8,10\right\}$.By a similar argument for other cases, i.e., $\beta_{K}\in \left\{4,8,10\right\}$, we always obtain a contradiction. Then, we finish the proof. □

The following result shows that Strichartz’s method (Theorem 2.8 in [24]) is not applicable to the above set Λ(N,B).

**Theorem** **4.**
*We have*

$$\underset{n\to \infty }{lim inf}\underset{\lambda \in \Lambda^{n}}{\inf}\left|m_{D}\left(N^{-n}\lambda \right)\right|=0.$$



**Proof.** Obviously, we need only to prove that there exists a subsequence ${\left\{\lambda_{n_{k}}\right\}}_{k\geq 1}\subset \Lambda^{n_{k}}$ such that $N^{-n_{k}}\lambda_{n_{k}}$ tends to a zero point of $m_{D}$ as *k* tends to infinity. Let $T_{0}$ be the attractor of the $IFS\{\Phi_{j}\left(x\right)=\frac{1}{6}\left(x+j\right):j\in {⋃}_{i=1}^{4}B_{i}\}$. Thus, we have $T_{0}\subset \left[0,\;\frac{52}{5}\right]$.For k≥M, we write $n_{k}=2^{2k+2}+2k+1$, and we take
$$\beta_{n_{k}}=38+52\times 6+38\times 6^{2}+52\times 6^{3}+\cdots +38\times 6^{2^{2k+1}}\in \Lambda_{2^{2k+1}+2k-1}^{n_{k}},$$
where the coefficients 38 and 52 appear alternately. By a simple deduction, we obtain
(41)$$6^{-2^{2k+1}-2}\left(10+\beta_{n_{k}}\right)=\frac{4}{3}.$$
Take arbitrarily $\alpha \in \Lambda^{2^{2k+1}+2k-1}$, and write
$$\lambda_{n_{k}}=\alpha +6^{2^{2k+1}+2k-1}\beta_{n_{k}}.$$
By (Equation 40), we obtain
$$\lambda_{n_{k}}\in \Lambda^{n_{k}}.$$
According to the definition of $T_{0}$, we have
$$6^{-2^{2k+1}-2k+1}\alpha \in T_{0},$$
which implies $|6^{-2^{2k+1}-2k+1}\alpha -10|\leq \frac{52}{5}$. In combination with (Equation 41), we have
$$\begin{array}{cc} \; & \left|6^{-n_{k}}\lambda_{n_{k}}-\frac{4}{3}\right|\\ = & \left|6^{-2^{2k+1}-2}\left(6^{-2^{2k+1}-2k+1}\alpha +\beta_{n_{k}}\right)-6^{-2^{2k+1}-2}\left(10+\beta_{n_{k}}\right)\right|\\ = & \left|6^{-2^{2k+1}-2}\left(6^{-2^{2k+1}-2k+1}\alpha -10\right)\right|\\ ⩽ & 6^{-2^{2k+1}-2}\times \frac{52}{5}. \end{array}$$
Noting the fact that $m_{D}\left(\frac{4}{3}\right)=0$, we finish the proof. □

## 5. Summary and Conclusions

In this paper, we introduced a tree structure with the language of symbolic space. The natural spectrum candidate of a self-similar measure associated with an IFS is a set with a special tree structure. We obtained three equivalent conclusions for Λ to be a spectrum of a self-similar measure. One of them implies that there exists an infinite orbit with an element of a nonzero integer associated with the dual IFS. An example involving a self-similar measure and a spectrum candidate $\Lambda \left(N,S\right)=S_{0}+NS_{1}+N^{2}S_{2}\cdots$ showed the tree structure expands essentially the field of spectrum candidates.

It is one of the most important problems to find all spectra of a spectral measure. We are not sure that every spectrum of a self-similar measure holds a tree structure. On the other hand, the self-similar $\mu_{N,D}$ measure has another description, $\mu_{N,D}=\delta_{\frac{1}{N}D}*\delta_{\frac{1}{N^{2}}D}*\cdots$. It is obvious to ask if Theorem 2 holds for the Moran-type self-similar measure. As mentioned in the Introduction, the version of Theorem 1 in higher-dimensional space has not been obtained completely. It is the next research direction to prove Theorem 2 the for self-affine measures.

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
