# Peer review of "Spectra of Self-Similar Measures"

_entropy, 2022, doi:10.3390/e24081142_

Round 1
Reviewer 1 Report
The report is attached.

Author Response
The main goal of this paper is to prove Theorem 3.1, which improved the work of Laba and Wang (2015). The process of proving this theorem confuses the reader, it is better to prove lemmas 3.2, 3.3 and 3.4 first, then prove theorem 3.1. Due to the large number of claims in the proof of Theorem 3.1, it is suggested to assign a label to each claim. As far as I am concerned,the spectral measures introduced in the article is of great significance. Theorem 3.1 is a new
result and play a pivotal role in the article. The authors have conducted a considerable in-depth study with respect to the considered research issues. And I am very pleased to recommend that the present article could be accepted for publication in the Journal \Entropy Journal" after a minor revision.
Response: The proof of lemma 3.3 bases on the proof of Theorem 3.1 iii=>i .We have assigned a label to each claim. Sorry for the confuses caused by the lack of label of claims.
Some minor corrections are listed in the following.
1. Page 2, Line(8), (see for example, [1, 2, 3, 4, 5, 6, 7, 8, 9, 10, 11, 12, 16, 17, 18, 19, 20, 21,
22] and the references therein). These are too much references without any details of them.The authors should emphasise about the importance and significance of such study or may give some better literature review.
Response 1: The middle part of the introduction has been rewritten.
2. Section 3, First line: In this section, we will give our main theorems and show them. Rewrite this line.
Response 2: The line is rewritten as “In this section, we will give our main results involving three equivalent statements. To prove the most difficult part of the proof, we prepare several lemmas including a new criterion for a spectrum candidate with tree structure to be a spectrum of a self-similar measure. In the end of this section, we show that the new criterion is just a sufficient and necessary condition, which is stated as a corollary .”
3. Page 9, Line(-3) "Theorem" should not be bold. Page 12, Line(-8) "The proof " should be bold.
Response 3: The bold style of "Theorem" in Line(-3) Page 9 has been deleted. "The proof"in Line (-8) Page 12 has been chosen a bold style.
4. Page 16, Line(+12) "Proof of Claim" should be bold or simple Proof.
Response 4. "Proof of Claim" has been modified as bold "Proof".
5. Page 19, Line(-6) and Line(-7) "Theorem" should not be bold.
Response 5: The bold style of "Theorem" in Line(-6) and Line (-7) Page 19 has been cancelled.
6. Page 20, Line(+13) Claim 1 does not exist. (Which claim?)
Response 6: The label of Claim 2 in Page 20 has been assigned.
7. Page 21, Line(+3) Claim 1 does not exist. (Which claim?)
Response 7: The label of Claim 2 in Page 20 has been assigned.
8. In example, You claim that : Neither the criterion of both Laba and Wang (Theorem 1.1) non that of Strichartz ([23]) are applicable to this set Λ. Give some explanation here.
Response 8: The explanation is in the end of the example.
9.. Page 22, Line(-7) A brief explanation is needed as to why the pair ( 1 6D; Bi) is a compatible pair.
Response 9: The sentence has been rewritten, which includes the explanation.
10. A dot should be placed at the end of references number 1, 2, 3, 6, 16, 18, 21 and 22. Moreover, in Reference 15 and 16 you write complete Journal name as Journal of Functional Analysis, and in other J. Funct. Anal.. There must be a uniform pattern.
Response 10: A dot has been placed at the end of reference number 1, 2, 3, 6, 16, 18, 21 and 22.
The journal name has been modified as a uniform pattern.
Reviewer 2 Report
The paper deals with spectral self-similar measures generated by finite integer digit sets. The authors give necessary and sufficient conditions that a spectrum candidate is the spectrum of a special spectral self-similar measure. The results of the paper are new and interesting. The article gives an important contribution to the theory. To the best of my understanding, the arguments are correct. I recommend that this article can be published, however, I have some suggestions:
1) At the end of page 2: Explain the extensions [11] and [20], and how their relationship with the main theorem of the paper;
2) The text needs careful revision, there are some typos, for example, on Page 3, line 3, "in other word" -> "in other words";
3) At the end of section 1: The authors could add a small paragraph explaining why the example in section 4 is important and interesting;
Author Response
1) At the end of page 2: Explain the extensions [11] and [20], and how their relationship with the main theorem of the paper;
Response 1: The explanation of the extensions [11] and [20] is in the line-3, page 3, which includes their relationship with our results.
2) The text needs careful revision, there are some typos, for example, on Page 3, line 3, "in other word" -> "in other words";
Response 2: The manuscript has been checked carefully again. Thanks!
3) At the end of section 1: The authors could add a small paragraph explaining why the example in section 4 is important and interesting;
Response 3: A paragraph has been added to the end of Section 1 to explain the example in Section 4.
Reviewer 3 Report
This paper is interesting, hence, I recommend this paper for publication after the following major revisions:
1-Please add a section for the conclusions and the future research direction must be shown in the conclusion.
2-The abstract should contain answers to the following questions: What problem was studied and why is it important? What methods were used? What are the important results? What conclusions can be drawn from the results? What is the novelty of the work and where does it go beyond previous efforts in the literature?
3-The section of the introduction needs to improve and write the aims for this paper and the previous studies.
4-The manuscript needs proofreading to improve its presentation.
5-They should double-check the mathematical formulations, and also add appropriate references for governing equations.
Author Response
1-Please add a section for the conclusions and the future research direction must be shown in the conclusion.
Response 1: A section "Summary and Conclusions" has been added to the manuscript.
2-The abstract should contain answers to the following questions: What problem was studied and why is it important? What methods were used? What are the important results? What conclusions can be drawn from the results? What is the novelty of the work and where does it go beyond previous efforts in the literature?
Response 2: The abstract has been rewritten.
3-The section of the introduction needs to improve and write the aims for this paper and the previous studies.
Response 3: The middle part of the introduction has been rewritten.
4-The manuscript needs proofreading to improve its presentation.
Response 4: The manuscript has been checked again. Thanks for your suggestions!
5-They should double-check the mathematical formulations, and also add appropriate references for governing equations.
Response 5: We have finished the proofreading.
Round 2
Reviewer 3 Report
I am satisfied with the changes made. Therefore, my suggestion is to recommend the manuscript for publication.